# Statistical Active Learning Algorithms

**Maria Florina Balcan**
Georgia Institute of Technology
ninamf@cc.gatech.edu

**Vitaly Feldman**
IBM Research - Almaden
vitaly@post.harvard.edu

## Abstract

We describe a framework for designing efficient active learning algorithms that are tolerant to random classification noise and differentially-private. The framework is based on active learning algorithms that are *statistical* in the sense that they rely on estimates of expectations of functions of filtered random examples. It builds on the powerful statistical query framework of Kearns [30].

We show that any efficient active statistical learning algorithm can be automatically converted to an efficient active learning algorithm which is tolerant to random classification noise as well as other forms of "uncorrelated" noise. We show that commonly studied concept classes including thresholds, rectangles, and linear separators can be efficiently actively learned in our framework. These results combined with our generic conversion lead to the first computationally-efficient algorithms for actively learning some of these concept classes in the presence of random classification noise that provide exponential improvement in the dependence on the error $\epsilon$ over their passive counterparts. In addition, we show that our algorithms can be automatically converted to efficient active differentially-private algorithms. This leads to the first differentially-private active learning algorithms with exponential label savings over the passive case.

## 1   Introduction

Most classic machine learning methods depend on the assumption that humans can annotate all the data available for training. However, many modern machine learning applications have massive amounts of unannotated or unlabeled data. As a consequence, there has been tremendous interest both in machine learning and its application areas in designing algorithms that most efficiently utilize the available data, while minimizing the need for human intervention. An extensively used and studied technique is active learning, where the algorithm is presented with a large pool of unlabeled examples and can interactively ask for the labels of examples of its own choosing from the pool, with the goal to drastically reduce labeling effort. This has been a major area of machine learning research in the past decade [19], with several exciting developments on understanding its underlying statistical principles [27, 18, 4, 3, 29, 21, 15, 7, 31, 10, 34, 6]. In particular, several general characterizations have been developed for describing when active learning can in principle have an advantage over the classic passive supervised learning paradigm, and by how much. However, these efforts were primarily focused on sample size bounds rather than computation, and as a result many of the proposed algorithms are not computationally efficient. The situation is even worse in the presence of noise where active learning appears to be particularly hard. In particular, prior to this work, there were no known efficient active algorithms for concept classes of super-constant VC-dimension that are provably robust to random and independent noise while giving improvements over the passive case.

**Our Results:**   We propose a framework for designing efficient (polynomial time) active learning algorithms which is based on restricting the way in which examples (both labeled and unlabeled) are accessed by the algorithm. These restricted algorithms can be easily simulated using active sampling and, in addition, possess a number of other useful properties. The main property we will consider is

tolerance to random classification noise of rate $\eta$ (each label is flipped randomly and independently with probability $\eta$ [1]). Further, as we will show, the algorithms are tolerant to other forms of noise and can be simulated in a differentially-private way.

In our restriction, instead of access to random examples from some distribution $P$ over $X \times Y$ the learning algorithm only gets "active" estimates of the statistical properties of $P$ in the following sense. The algorithm can choose any *filter* function $\chi(x) : X \to [0,1]$ and a query function $\phi : X \times Y \to [-1,1]$ for any $\chi$ and $\phi$. For simplicity we can think of $\chi$ as an indicator function of some set $\chi_S \subseteq X$ of "informative" points and of $\phi$ as some useful property of the target function. For this pair of functions the learning algorithm can get an estimate of $\mathbf{E}_{(x,y)\sim P}[\phi(x,y) \mid x \in \chi_S]$. For $\tau$ and $\tau_0$ chosen by the algorithm the estimate is provided to within *tolerance* $\tau$ as long as $\mathbf{E}_{(x,y)\sim P}[x \in \chi_S] \geq \tau_0$ (nothing is guaranteed otherwise). Here the inverse of $\tau$ corresponds to the label complexity of the algorithm and the inverse of $\tau_0$ corresponds to its unlabeled sample complexity. Such a query is referred to as *active statistical query (SQ)* and algorithms using active SQs are referred to as *active statistical algorithms*.

Our framework builds on the classic statistical query (SQ) learning framework of Kearns [30] defined in the context of PAC learning model [35]. The SQ model is based on estimates of expectations of functions of examples (but without the additional filter function) and was defined in order to design efficient noise tolerant algorithms in the PAC model. Despite the restrictive form, most of the learning algorithms in the PAC model and other standard techniques in machine learning and statistics used for problems over distributions have SQ analogues [30, 12, 11, **?**][1]. Further, statistical algorithms enjoy additional properties: they can be simulated in a differentially-private way [11], automatically parallelized on multi-core architectures [17] and have known information-theoretic characterizations of query complexity [13, 26]. As we show, our framework inherits the strengths of the SQ model while, as we will argue, capturing the power of active learning.

At a first glance being active and statistical appear to be incompatible requirements on the algorithm. Active algorithms typically make label query decisions on the basis of examining individual samples (for example as in binary search for learning a threshold or the algorithms in [27, 21, 22]). At the same time statistical algorithms can only examine properties of the underlying distribution. But there also exist a number of active learning algorithms that can be seen as applying passive learning techniques to batches of examples that are obtained from querying labels of samples that satisfy the same filter. These include the general $A^2$ algorithm [4] and, for example, algorithms in [3, 20, 9, 8]. As we show, we can build on these techniques to provide algorithms that fit our framework.

We start by presenting a general reduction showing that any efficient active statistical learning algorithm can be automatically converted to an efficient active learning algorithm which is tolerant to random classification noise as well as other forms of "uncorrelated" noise. We then demonstrate the generality of our framework by showing that the most commonly studied concept classes including thresholds, balanced rectangles, and homogenous linear separators can be efficiently actively learned via active statistical algorithms. For these concept classes, we design efficient active learning algorithms that are statistical and provide the same exponential improvements in the dependence on the error $\epsilon$ over passive learning as their non-statistical counterparts.

The primary problem we consider is active learning of homogeneous halfspaces, a problem that has attracted a lot of interest in the theory of active learning [27, 18, 3, 9, 22, 16, 23, 8, 28]. We describe two algorithms for the problem. First, building on insights from margin based analysis [3, 8], we give an active statistical learning algorithm for homogeneous halfspaces over all isotropic log-concave distributions, a wide class of distributions that includes many well-studied density functions and has played an important role in several areas including sampling, optimization, and learning [32]. Our algorithm for this setting proceeds in rounds; in round $t$ we build a better approximation $w_t$ to the target function by using a passive SQ learning algorithm (e.g., the one of [24]) over a distribution $D_t$ that is a mixture of distributions in which each component is the original distribution conditioned on being within a certain distance from the hyperplane defined by previous approximations $w_i$. To perform passive statistical queries relative to $D_t$ we use active SQs with a corresponding real valued filter. This algorithm is computationally efficient and uses only $\text{poly}(d, \log(1/\epsilon))$ active statistical queries of tolerance inverse-polynomial in the dimension $d$ and $\log(1/\epsilon)$.

For the special case of the uniform distribution over the unit ball we give a new, simpler and substantially more efficient active statistical learning algorithm. Our algorithm is based on measuring the error of a halfspace conditioned on being within some margin of that halfspace. We show that such measurements performed on the perturbations of the current hypothesis along the $d$ basis vectors can be combined to derive a better hypothesis. This approach differs substantially from the previous algorithms for this problem [3, 22]. The algorithm is computationally efficient and uses $d \log(1/\epsilon)$ active SQs with tolerance of $\Omega(1/\sqrt{d})$ and filter tolerance of $\Omega(1/\epsilon)$.

These results, combined with our generic simulation of active statistical algorithms in the presence of random classification noise (RCN) lead to the first known computationally efficient algorithms for actively learning halfspaces which are RCN tolerant and give provable label savings over the passive case. For the uniform distribution case this leads to an algorithm with sample complexity of $O((1 - 2\eta)^{-2} \cdot d^2 \log(1/\epsilon) \log(d \log(1/\epsilon)))$ and for the general isotropic log-concave case we get sample complexity of $\text{poly}(d, \log(1/\epsilon), 1/(1 - 2\eta))$. This is worse than the sample complexity in the noiseless case which is just $O((d + \log \log(1/\epsilon)) \log(1/\epsilon))$ [8]. However, compared to passive learning in the presence of RCN, our algorithms have exponentially better dependence on $\epsilon$ and essentially the same dependence on $d$ and $1/(1 - 2\eta)$. One issue with the generic simulation is that it requires knowledge of $\eta$ (or an almost precise estimate). Standard approach to dealing with this issue does not always work in the active setting and for our log-concave and the uniform distribution algorithms we give a specialized argument that preserves the exponential improvement in the dependence on $\epsilon$.

**Differentially-private active learning:** In many application of machine learning such as medical and financial record analysis, data is both sensitive and expensive to label. However, to the best of our knowledge, there are no formal results addressing both of these constraints. We address the problem by defining a natural model of differentially-private active learning. In our model we assume that a learner has full access to unlabeled portion of some database of $n$ examples $S \subseteq X \times Y$ which correspond to records of individual participants in the database. In addition, for every element of the database $S$ the learner can request the label of that element. As usual, the goal is to minimize the number of label requests (such setup is referred to as *pool-based* active learning [33]). In addition, we would like to preserve the *differential privacy* of the participants in the database, a now-standard notion of privacy introduced in [25]. Informally speaking, an algorithm is differentially private if adding any record to $S$ (or removing a record from $S$) does not affect the probability that any specific hypothesis will be output by the algorithm significantly.

As first shown by [11], SQ algorithms can be automatically translated into differentially-private algorithms. Using a similar approach, we show that active SQ learning algorithms can be automatically transformed into differentially-private active learning algorithms. Using our active statistical algorithms for halfspaces we obtain the first algorithms that are both differentially-private and give exponential improvements in the dependence of label complexity on the accuracy parameter $\epsilon$.

**Additional related work:** As we have mentioned, most prior theoretical work on active learning focuses on either sample complexity bounds (without regard for efficiency) or the noiseless case. For random classification noise in particular, [6] provides a sample complexity analysis based on the splitting index that is optimal up to polylog factors and works for general concept classes and distributions, but it is not computationally efficient. In addition, several works give active learning algorithms with empirical evidence of robustness to certain types of noise [9, 28];

In [16, 23] online learning algorithms in the selective sampling framework are presented, where labels must be actively queried before they are revealed. Under the assumption that the label conditional distribution is a linear function determined by a fixed target vector, they provide bounds on the regret of the algorithm and on the number of labels it queries when faced with an adaptive adversarial strategy of generating the instances. As pointed out in [23], these results can also be converted to a distributional PAC setting where instances $x_t$ are drawn i.i.d. In this setting they obtain exponential improvement in label complexity over passive learning. These interesting results and techniques are not directly comparable to ours. Our framework is not restricted to halfspaces. Another important difference is that (as pointed out in [28]) the exponential improvement they give is not possible in the noiseless version of their setting. In other words, the addition of linear noise defined by the target makes the problem easier for active sampling. By contrast RCN can only make the classification task harder than in the realizable case.

Due to space constraint details of most proofs and further discussion appear in the full version of this paper [5].

## 2 Active Statistical Algorithms

Let $X$ be a domain and $P$ be a distribution over labeled examples on $X$. We represent such a distribution by a pair $(D, \psi)$ where $D$ is the marginal distribution of $P$ on $X$ and $\psi : X \to [-1, 1]$ is a function defined as $\psi(z) = \mathbf{E}_{(x,\ell) \sim P}[\ell \mid x = z]$. We will be primarily considering learning in the PAC model (realizable case) where $\psi$ is a boolean function, possibly corrupted by random noise.

When learning with respect to a distribution $P = (D, \psi)$, an active statistical learner has access to *active statistical queries*. A query of this type is a pair of functions $(\chi, \phi)$, where $\chi : X \to [0, 1]$ is the *filter* function which for a point $x$, specifies the probability with which the label of $x$ should be queried. The function $\phi : X \times \{-1, 1\} \to [-1, 1]$ is the query function and depends on both point and the label. The filter function $\chi$ defines the distribution $D$ conditioned on $\chi$ as follows: for each $x$ the density function $D_{|\chi}(x)$ is defined as $D_{|\chi}(x) = D(x)\chi(x)/\mathbf{E}_D[\chi(x)]$. Note that if $\chi$ is an indicator function of some set $S$ then $D_{|\chi}$ is exactly $D$ conditioned on $x$ being in $S$. Let $P_{|\chi}$ denote the conditioned distribution $(D_{|\chi}, \psi)$. In addition, a query has two tolerance parameters: filter tolerance $\tau_0$ and query tolerance $\tau$. In response to such a query the algorithm obtains a value $\mu$ such that if $\mathbf{E}_D[\chi(x)] \geq \tau_0$ then

$$\left| \mu - \mathbf{E}_{P_{|\chi}}[\phi(x, \ell)] \right| \leq \tau$$

(and nothing is guaranteed when $\mathbf{E}_D[\chi(x)] < \tau_0$).

An active statistical learning algorithm can also ask *target-independent* queries with tolerance $\tau$ which are just queries over unlabeled samples. That is for a query $\varphi : X \to [-1, 1]$ the algorithm obtains a value $\mu$, such that $|\mu - \mathbf{E}_D[\varphi(x)]| \leq \tau$. Such queries are not necessary when $D$ is known to the learner. Also for the purposes of obtaining noise tolerant algorithms one can relax the requirements of model and give the learning algorithm access to unlabelled samples.

Our definition generalizes the statistical query framework of Kearns [30] which does not include filtering function, in other words a query is just a function $\phi : X \times \{-1, 1\} \to [-1, 1]$ and it has a single tolerance parameter $\tau$. By definition, an active SQ $(\chi, \phi)$ with tolerance $\tau$ relative to $P$ is the same as a passive statistical query $\phi$ with tolerance $\tau$ relative to the distribution $P_{|\chi}$. In particular, a (passive) SQ is equivalent to an active SQ with filter $\chi \equiv 1$ and filter tolerance 1.

We note that from the definition of active SQ we can see that

$$\mathbf{E}_{P_{|\chi}}[\phi(x, \ell)] = \mathbf{E}_P[\phi(x, \ell) \cdot \chi(x)]/\mathbf{E}_P[\chi(x)].$$

This implies that an active statistical query can be estimated using two passive statistical queries. However to estimate $\mathbf{E}_{P_{|\chi}}[\phi(x, \ell)]$ with tolerance $\tau$ one needs to estimate $\mathbf{E}_P[\phi(x, \ell) \cdot \chi(x)]$ with tolerance $\tau \cdot \mathbf{E}_P[\chi(x)]$ which can be much lower than $\tau$. Tolerance of a SQ directly corresponds to the number of examples needed to evaluate it and therefore simulating active SQs passively might require many more labeled examples.

### 2.1 Simulating Active Statistical Queries

We first note that a valid response to a target-independent query with tolerance $\tau$ can be obtained, with probability at least $1 - \delta$, using $O(\tau^{-2} \log(1/\delta))$ unlabeled samples.

A natural way of simulating an active SQ is by filtering points drawn randomly from $D$: draw a random point $x$, let $B$ be drawn from Bernoulli distribution with probability of success $\chi(x)$; ask for the label of $x$ when $B = 1$. The points for which we ask for a label are distributed according to $D_{|\chi}$. This implies that the empirical average of $\phi(x, \ell)$ on $O(\tau^{-2} \log(1/\delta))$ labeled examples will then give $\mu$. Formally we get the following theorem.

**Theorem 2.1.** *Let $P = (D, \psi)$ be a distribution over $X \times \{-1, 1\}$. There exists an active sampling algorithm that given functions $\chi : X \to [0, 1]$, $\phi : X \times \{-1, 1\} \to [-1, 1]$, values $\tau_0 > 0$, $\tau > 0$, $\delta > 0$, and access to samples from $P$, with probability at least $1 - \delta$, outputs a valid response to active statistical query $(\chi, \phi)$ with tolerance parameters $(\tau_0, \tau)$. The algorithm uses $O(\tau^{-2} \log(1/\delta))$ labeled examples from $P$ and $O(\tau_0^{-1} \tau^{-2} \log(1/\delta))$ unlabeled samples from $D$.*

A direct way to simulate all the queries of an active SQ algorithm is to estimate the response to each query using fresh samples and use the union bound to ensure that, with probability at least $1 - \delta$, all queries are answered correctly. Such direct simulation of an algorithm that uses at most $q$ queries can be done using $O(q\tau^{-2}\log(q/\delta))$ labeled examples and $O(q\tau_0^{-1}\tau^{-2}\log(q/\delta))$ unlabeled samples. However in many cases a more careful analysis can be used to reduce the sample complexity of simulation.

Labeled examples can be shared to simulate queries that use the same filter $\chi$ and do not depend on each other. This implies that the sample size sufficient for simulating $q$ non-adaptive queries with the same filter scales logarithmically with $q$. More generally, given a set of $q$ query functions (possibly chosen adaptively) which belong to some set $Q$ of low complexity (such as VC dimension) one can reduce the sample complexity of estimating the answers to all $q$ queries (with the same filter) by invoking the standard bounds based on uniform convergence (e.g. [14]).

## 2.2 Noise tolerance

An important property of the simulation described in Theorem 2.1 is that it can be easily adapted to the case when the labels are corrupted by random classification noise [1]. For a distribution $P = (D, \psi)$ let $P^\eta$ denote the distribution $P$ with the label flipped with probability $\eta$ randomly and independently of an example. It is easy to see that $P^\eta = (D, (1 - 2\eta)\psi)$. We now show that, as in the SQ model [30], active statistical queries can be simulated given examples from $P^\eta$.

**Theorem 2.2.** *Let $P = (D, \psi)$ be a distribution over examples and let $\eta \in [0, 1/2)$ be a noise rate. There exists an active sampling algorithm that given functions $\chi : X \to [0, 1]$, $\phi : X \times \{-1, 1\} \to [-1, 1]$, values $\eta$, $\tau_0 > 0$, $\tau > 0$, $\delta > 0$, and access to samples from $P^\eta$, with probability at least $1 - \delta$, outputs a valid response to active statistical query $(\chi, \phi)$ with tolerance parameters $(\tau_0, \tau)$. The algorithm uses $O(\tau^{-2}(1 - 2\eta)^{-2}\log(1/\delta))$ labeled examples from $P^\eta$ and $O(\tau_0^{-1}\tau^{-2}(1 - 2\eta)^{-2}\log(1/\delta))$ unlabeled samples from $D$.*

Note that the sample complexity of the resulting active sampling algorithm has information-theoretically optimal quadratic dependence on $1/(1 - 2\eta)$, where $\eta$ is the noise rate.

**Remark 2.3.** *This simulation assumes that $\eta$ is given to the algorithm exactly. It is easy to see from the proof, that any value $\eta'$ such that $\frac{1-2\eta}{1-2\eta'} \in [1 - \tau/4, 1 + \tau/4]$ can be used in place of $\eta$ (with the tolerance of estimating $\mathbf{E}_{P_{|\chi}^\eta}[\frac{1}{2}(\phi(x, 1) - \phi(x, -1)) \cdot \ell]$ set to $(1 - 2\eta)\tau/4$). In some learning scenarios even an approximate value of $\eta$ is not known but it is known that $\eta \le \eta_0 < 1/2$. To address this issue one can construct a sequence $\eta_1, \ldots, \eta_k$ of guesses of $\eta$, run the learning algorithm with each of those guesses in place of the true $\eta$ and let $h_1, \ldots, h_k$ be the resulting hypotheses [30]. One can then return the hypothesis $h_i$ among those that has the best agreement with a suitably large sample. It is not hard to see that $k = O(\tau^{-1} \cdot \log(1/(1 - 2\eta_0)))$ guesses will suffice for this strategy to work [2].*

*Passive hypothesis testing requires $\Omega(1/\epsilon)$ labeled examples and might be too expensive to be used with active learning algorithms. It is unclear if there exists a general approach for dealing with unknown $\eta$ in the active learning setting that does not increase substantially the labelled example complexity. However, as we will demonstrate, in the context of specific active learning algorithms variants of this approach can be used to solve the problem.*

We now show that more general types of noise can be tolerated as long as they are "uncorrelated" with the queries and the target function. Namely, we represent label noise using a function $\Lambda : X \to [0, 1]$, where $\Lambda(x)$ gives the probability that the label of $x$ is flipped. The rate of $\Lambda$ when learning with respect to marginal distribution $D$ over $X$ is $\mathbf{E}_D[\Lambda(x)]$. For a distribution $P = (D, \psi)$ over examples, we denote by $P^\Lambda$ the distribution $P$ corrupted by label noise $\Lambda$. It is easy to see that $P^\Lambda = (D, \psi \cdot (1 - 2\Lambda))$. Intuitively, $\Lambda$ is "uncorrelated" with a query if the way that $\Lambda$ deviates from its rate is almost orthogonal to the query on the target distribution.

**Definition 2.4.** *Let $P = (D, \psi)$ be a distribution over examples and $\tau' > 0$. For functions $\chi : X \to [0, 1]$, $\phi : X \times \{-1, 1\} \to [-1, 1]$, we say that a noise function $\Lambda : X \to [0, 1]$ is $(\eta, \tau')$-uncorrelated with $\phi$ and $\chi$ over $P$ if,*

$$\left| \mathbf{E}_{D_{|\chi}} \left[ \frac{\phi(x, 1) - \phi(x, -1)}{2} \psi(x) \cdot (1 - 2(\Lambda(x) - \eta)) \right] \right| \le \tau'.$$

In this definition $(1 - 2(\Lambda(x) - \eta))$ is the expectation of $\{-1, 1\}$ coin that is flipped with probability $\Lambda(x) - \eta$, whereas $(\phi(x, 1) - \phi(x, -1))\psi(x)$ is the part of the query which measures the correlation with the label. We now give an analogue of Theorem 2.2 for this more general setting.

**Theorem 2.5.** *Let $P = (D, \psi)$ be a distribution over examples, $\chi : X \to [0, 1]$, $\phi : X \times \{-1, 1\} \to [-1, 1]$ be a query and a filter functions, $\eta \in [0, 1/2)$, $\tau > 0$ and $\Lambda$ be a noise function that is $(\eta, (1 - 2\eta)\tau/4)$-uncorrelated with $\phi$ and $\chi$ over $P$. There exists an active sampling algorithm that given functions $\chi$ and $\phi$, values $\eta$, $\tau_0 > 0$, $\tau > 0$, $\delta > 0$, and access to samples from $P^\Lambda$, with probability at least $1 - \delta$, outputs a valid response to active statistical query $(\chi, \phi)$ with tolerance parameters $(\tau_0, \tau)$. The algorithm uses $O(\tau^{-2}(1 - 2\eta)^{-2} \log(1/\delta))$ labeled examples from $P^\Lambda$ and $O(\tau_0^{-1} \tau^{-2}(1 - 2\eta)^{-2} \log(1/\delta))$ unlabeled samples from $D$.*

An immediate implication of Theorem 2.5 is that one can simulate an active SQ algorithm $A$ using examples corrupted by noise $\Lambda$ as long as $\Lambda$ is $(\eta, (1 - 2\eta)\tau/4)$-uncorrelated with all $A$'s queries of tolerance $\tau$ for some fixed $\eta$.

## 2.3 Simple examples

**Thresholds:** We show that a classic example of active learning a threshold function on an interval can be easily expressed using active SQs. For simplicity and without loss of generality we can assume that the interval is $[0, 1]$ and the distribution is uniform over it (as usual, we can bring the distribution to be close enough to this form using unlabeled samples or target-independent queries). Assume that we know that the threshold $\theta$ belongs to the interval $[a, b] \subseteq [0, 1]$. We ask a query $\phi(x, \ell) = (\ell + 1)/2$ with filter $\chi(x)$ which is the indicator function of the interval $[a, b]$ with tolerance $1/4$ and filter tolerance $b - a$. Let $v$ be the response to the query. By definition, $\mathbf{E}[\chi(x)] = b - a$ and therefore we have that $|v - \mathbf{E}[\phi(x, \ell) \mid x \in [a, b]]| \leq 1/4$. Note that,

$$\mathbf{E}[\phi(x, \ell) \mid x \in [a, b]] = (b - \theta)/(b - a) .$$

We can therefore conclude that $(b - \theta)/(b - a) \in [v - 1/4, v + 1/4]$ which means that $\theta \in [b - (v + 1/4)(b - a), b - (v - 1/4)(b - a)] \cap [a, b]$. Note that the length of this interval is at most $(b - a)/2$. This means that after at most $\log_2(1/\epsilon) + 1$ iterations we will reach an interval $[a, b]$ of length at most $\epsilon$. In each iteration only constant $1/4$ tolerance is necessary and filter tolerance is never below $\epsilon$. A direct simulation of this algorithm can be done using $\log(1/\epsilon) \cdot \log(\log(1/\epsilon)/\delta)$ labeled examples and $\tilde{O}(1/\epsilon) \cdot \log(1/\delta)$ unlabeled samples.

Learning of thresholds can also be easily used to obtain a simple algorithm for learning axis-aligned rectangles whose weight under the target distribution is not too small.

$A^2$ : We now note that the general and well-studied $A^2$ algorithm of [4] falls naturally into our framework. At a high level, the $A^2$ algorithm is an iterative, *disagreement-based* active learning algorithm. It maintains a set of surviving classifiers $C_i \subseteq C$, and in each round the algorithm asks for the labels of a few random points that fall in the current region of disagreement of the surviving classifiers. Formally, the region of disagreement $\text{DIS}(C_i)$ of a set of classifiers $C_i$ is the of set of instances $x$ such that for each $x \in \text{DIS}(C_i)$ there exist two classifiers $f, g \in C_i$ that disagree about the label of $x$. Based on the queried labels, the algorithm then eliminates hypotheses that were still under consideration, but only if it is *statistically confident* (given the labels queried in the last round) that they are suboptimal. In essence, in each round $A^2$ only needs to estimate the error rates (of hypotheses still under consideration) under the conditional distribution of being in the region of disagreement. This can be easily done via active statistical queries. Note that while the number of active statistical queries needed to do this could be large, the number of labeled examples needed to simulate these queries is essentially the same as the number of labeled examples needed by the known $A^2$ analyses [29]. While in general the required computation of the disagreement region and manipulations of the hypothesis space cannot be done efficiently, efficient implementation is possible in a number of simple cases such as when the VC dimension of the concept class is a constant. It is not hard to see that in these cases the implementation can also be done using a statistical algorithm.

# 3 Learning of halfspaces

In this section we outline our reduction from active learning to passive learning of homogeneous linear separators based on the analysis of Balcan and Long [8]. Combining it with the SQ learning algorithm for halfspaces by Dunagan and Vempala [24], we obtain the first efficient noise-tolerant active learning of homogeneous halfspaces for any isotropic log-concave distribution. One of the key point of this result is that it is relatively easy to harness the involved results developed for SQ framework to obtain new active statistical algorithms.

Let $\mathcal{H}_d$ denote the concept class of all homogeneous halfspaces. Recall that a distribution over $\mathbb{R}^d$ is log-concave if $\log f(\cdot)$ is concave, where $f$ is its associated density function. It is isotropic if its mean is the origin and its covariance matrix is the identity. Log-concave distributions form a broad class of distributions: for example, the Gaussian, Logistic, Exponential, and uniform distribution over any convex set are log-concave distributions. Using results in [24] and properties of log-concave distributions, we can show:

**Theorem 3.1.** *There exists a SQ algorithm* LearnHS *that learns* $\mathcal{H}_d$ *to accuracy* $1 - \epsilon$ *over any distribution* $D_{|\chi}$*, where* $D$ *is an isotropic log-concave distribution and* $\chi : \mathbb{R}^d \to [0, 1]$ *is a filter function. Further* LearnHS *outputs a homogeneous halfspace, runs in time polynomial in* $d, 1/\epsilon$ *and* $\log(1/\lambda)$ *and uses SQs of tolerance* $\geq 1/poly(d, 1/\epsilon, \log(1/\lambda))$*, where* $\lambda = \mathbf{E}_D[\chi(x)]$*.*

We now state the properties of our new algorithm formally.

**Theorem 3.2.** *There exists an active SQ algorithm* ActiveLearnHS-LogC *(Algorithm 1) that for any isotropic log-concave distribution* $D$ *on* $\mathbb{R}^d$*, learns* $\mathcal{H}_d$ *over* $D$ *to accuracy* $1 - \epsilon$ *in time* $poly(d, \log(1/\epsilon))$ *and using active SQs of tolerance* $\geq 1/poly(d, \log(1/\epsilon))$ *and filter tolerance* $\Omega(\epsilon)$*.*

---

**Algorithm 1** ActiveLearnHS-LogC: Active SQ learning of homogeneous halfspaces over isotropic log-concave densities

---

1: %% Constants $c$, $C_1$, $C_2$ and $C_3$ are determined by the analysis.
2: Run LearnHS with error $C_2$ to obtain $w_0$.
3: **for** $k = 1$ to $s = \lceil \log_2(1/(c\epsilon)) \rceil$ **do**
4:     Let $b_{k-1} = C_1/2^{k-1}$
5:     Let $\mu_k$ equal the indicator function of being within margin $b_{k-1}$ of $w_{k-1}$
6:     Let $\chi_k = (\sum_{i \leq k} \mu_i)/k$
7:     Run LearnHS over $D_k = D_{|\chi_k}$ with error $C_2/k$ by using active queries with filter $\chi_k$ and filter tolerance $C_3\epsilon$ to obtain $w_k$
8: **end for**
9: **return** $w_s$

---

We remark that, as usual, we can first bring the distribution to an isotropic position by using target independent queries to estimate the mean and the covariance matrix of the distribution. Therefore our algorithm can be used to learn halfspaces over general log-concave densities as long as the target halfspace passes through the mean of the density.

We can now apply Theorem 2.2 (or more generally Theorem 2.5) to obtain an efficient active learning algorithm for homogeneous halfspaces over log-concave densities in the presence of random classification noise of known rate. Further since our algorithm relies on LearnHS which can also be simulated when the noise rate is unknown (see Remark 2.3) we obtain an active algorithm which does not require the knowledge of the noise rate.

**Corollary 3.3.** *There exists a polynomial-time active learning algorithm that for any* $\eta \in [0, 1/2)$*, learns* $\mathcal{H}_d$ *over any log-concave distributions with random classification noise of rate* $\eta$ *to error* $\epsilon$ *using* $poly(d, \log(1/\epsilon), 1/(1 - 2\eta))$ *labeled examples and a polynomial number of unlabeled samples.*

For the special case of the uniform distribution on the unit sphere (or, equivalently for our purposes, unit ball) we give a substantially simpler and more efficient algorithm in terms of both sample and computational complexity. This setting was previously studied in [3, 22]. The detailed presentation of the technical ideas appears in the full version of the paper [5].

**Theorem 3.4.** *There exists an active SQ algorithm* `ActiveLearnHS-U` *that learns $\mathcal{H}_d$ over the uniform distribution on the $(d-1)$-dimensional unit sphere to accuracy $1 - \epsilon$, uses $(d + 1)\log(1/\epsilon)$ active SQs with tolerance of $\Omega(1/\sqrt{d})$ and filter tolerance of $\Omega(1/\epsilon)$ and runs in time $d \cdot poly(\log(d/\epsilon))$.*

# 4   Differentially-private active learning

In this section we show that active SQ learning algorithms can also be used to obtain differentially private active learning algorithms. Formally, for some domain $X \times Y$, we will call $S \subseteq X \times Y$ a *database*. Databases $S, S' \subset X \times Y$ are *adjacent* if one can be obtained from the other by modifying a single element. Here we will always have $Y = \{-1, 1\}$. In the following, $A$ is an algorithm that takes as input a database $D$ and outputs an element of some finite set $R$.

**Definition 4.1** (Differential privacy [25])**.** *A (randomized) algorithm $A : 2^{X \times Y} \to R$ is $\alpha$-differentially private if for all $r \in R$ and every pair of adjacent databases $S, S'$, we have $\Pr[A(S) = r] \le e^\epsilon \Pr[A(S') = r]$.*

Here we consider algorithms that operate on $S$ in an active way. That is the learning algorithm receives the unlabeled part of each point in $S$ as an input and can only obtain the label of a point upon request. The total number of requests is the label complexity of the algorithm.

**Theorem 4.2.** *Let $A$ be an algorithm that learns a class of functions $H$ to accuracy $1 - \epsilon$ over distribution $D$ using $M_1$ active SQs of tolerance $\tau$ and filter tolerance $\tau_0$ and $M_2$ target-independent queries of tolerance $\tau_u$. There exists a learning algorithm $A'$ that given $\alpha > 0, \delta > 0$ and active access to database $S \subseteq X \times \{-1, 1\}$ is $\alpha$-differentially private and uses at most $O([\frac{M_1}{\alpha\tau} + \frac{M_1}{\tau^2}]\log(M_1/\delta))$ labels. Further, for some $n = O([\frac{M_1}{\alpha\tau_0\tau} + \frac{M_1}{\tau_0\tau^2} + \frac{M_2}{\alpha\tau_u} + \frac{M_2}{\tau_u^2}]\log((M_1 + M_2)/\delta))$, if $S$ consists of at least $n$ examples drawn randomly from $D$ then with, probability at least $1 - \delta$, $A'$ outputs a hypothesis with accuracy $\ge 1 - \epsilon$ (relative to distribution $D$). The running time of $A'$ is the same as the running time of $A$ plus $O(n)$.*

An immediate consequence of Theorem 4.2 is that for learning of homogeneous halfspaces over uniform or log-concave distributions we can obtain differential privacy while essentially preserving the label complexity. For example, by combining Theorems 4.2 and 3.4, we can efficiently and differentially-privately learn homogeneous halfspaces under the uniform distribution with privacy parameter $\alpha$ and error parameter $\epsilon$ by using only $O(d\sqrt{d}\log(1/\epsilon))/\alpha + d^2\log(1/\epsilon))$ labels. However, it is known that any passive learning algorithm, even ignoring privacy considerations and noise requires $\Omega\left(\frac{d}{\epsilon} + \frac{1}{\epsilon}\log\left(\frac{1}{\delta}\right)\right)$ labeled examples. So for $\alpha \ge 1/\sqrt{d}$ and small enough $\epsilon$ we get better label complexity.

# 5   Discussion

Our work suggests that, as in passive learning, active statistical algorithms might be essentially as powerful as example-based efficient active learning algorithms. It would be interesting to find more general evidence supporting this claim or, alternatively, a counterexample. A nice aspect of (passive) statistical learning algorithms is that it is possible to prove unconditional lower bounds on such algorithms using SQ dimension [13] and its extensions. It would be interesting to develop an active analogue of these techniques and give meaningful lower bounds based on them.

**Acknowledgments** We thank Avrim Blum and Santosh Vempala for useful discussions. This work was supported in part by NSF grants CCF-0953192 and CCF-1101215, AFOSR grant FA9550-09-1-0538, ONR grant N00014-09-1-0751, and a Microsoft Research Faculty Fellowship.

## Footnotes

[1]The sample complexity of the SQ analogues might increase sometimes though.

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
