[Reviews · NeurIPS 2013]

Submitted by Assigned_Reviewer_2

The authors propose a class of active learning algorithms, termed 'active statistical' algorithms that is supposed to combine the noise robustness, and differential privacy of the standard statistical query (SQ) framework, with the active learning setting in which one wishes to minimizes the number of labels one requires by actively (adaptively) selecting which data points to label.

As defined, the active SQ algorithms asks queries that consist of two parts: a 'filter' f:X->[0,1] representing the probability that a point X should be included in the current query, and the function g:X x {-1,1}->[-1,1], which is a function of a point-label pair. Additionally, as part of the query, there are two tolerance parameters, t1,t2, and the guarantee is that provided the measure of examples defined by f is at least t1, one receives a noisy estimate of E[g(x,y)} to within t2, where the expectation is defined over the measure induced by f.

In general, I doubt this new model will be as helpful, or have as much mathematical structure as the general SQ model. That said, it does seem to be `the right' model for capturing noise tolerance in an active learning setting (which yields differential privacy), and the algorithmic results, while not earth-shattering, provide a nice illustration of the power of this model.

It would be nice to include a little more discussion of the relation between the general SQ framework and the proposed model--specifically, the fact that the proposed model of queries can be implemented in the SQ framework, just with a tolerance that scales according to the measure of the portion of the distribution in consideration for each query.
Summary: solid paper, with the `right' formalization of a SQ framework for active learning. The paper would be more enjoyable to read if it were less aggressively sold. I'm also not entirely sure how much interest this will have within the NIPS community....

Submitted by Assigned_Reviewer_6

Overall, the paper is well-written. A few minor points:

1. In line 182: you probably can allow target-independent queries as
having tolerance \tau_0 (rather than \tau).
2. Line 211: in many **cases** (instead of case)
3. In the definition of P^{\eta}, say that (1 - 2\eta) \psi should be
interpretted as a (randomised) boolean function.

Summary: This paper introduces an SQ version of the popular active learning
framework. The advantage of this formulation is that any algorithm
designed to use only statistical queries (new kind introduced) is
automatically robust to random classification noise. In addition, the
authors discuss algorithms for homogeneous linear separators under
log-concave and uniform distributions, simple algorithms for intervals and
rectangles, and implications for differentially private (active) learning.

Submitted by Assigned_Reviewer_8

The paper proposes a method for turning certain active learning algorithms into ative learning statistical queries (SQ) algorithm.
The main advantage of such a transformation is that SQ algorithms have certain noise tolerance merits.
The paper starts witha definition of active SQ queries, although the definition is rather intricate and not easy to parse, it does seem to make sense and it meets teh two requirements of SQ queries - it can be estimated from a bounded number of standard label queries (plus access to unlabeled samples) and it retains the essence of the noise tolerance of regular SQ queries.

The authors then describe how several known successful active learning algorithms can be transformed into the SQ active learning model. The more interesting of these is and algorithm for noise tolerant active learning of homogeneous half-spaces under isotropic log-concave distributions (that combines ideas from the recent work of Balcan and Long on the active learning of such classes with teh work of Dunagan and Vempala on learning half-spaces in the usual SQ model).

This is a solid paper with some meaningful but not surprising contributions. My main criticism about this paper is that it tends to oversell itself....
Summary: Nice solid paper, marrying the SQ model with Active Leaning.
Author Feedback

Author rebuttal: Assigned_Reviewer_2 =======
This response refers to the initial review. The new review differs substantially from the original one and corrects significant misunderstandings.
---
We would like to correct a major misunderstanding in this review. The following claim by the reviewer demonstrates the misunderstanding:

"In general, I am not convinced that this new model is especially fundamental, novel, or helpful. While the authors claim this to be a generalization of the standard SQ model, the opposite can also be claimed [...] Specifically, to implement an 'active statistical' query, f,g, one can simply query a standard SQ oracle with the function h(x,y)=g(x,y)*f(x), and then divide by the measure induced by f, namely the response to the SQ oracle f(x)."

What the reviewer has overlooked however is that such classic SQ implementation requires label-dependent queries of much higher tolerance. Queries of higher tolerance require more labeled examples to implement. Basically, this argument misses the main point of active learning: sure if one has no constraints on the number of labels to query then one could simulate any active learning algorithm using passively labeled examples.

More formally, we can implement an active statistical query given by filter f(x) and query function g(x,y) (with f(x) being boolean for simplicity) by making the standard statistical query for the E[g(x,y)*f(x)] and then dividing the result by E[f(x)] (which can also be obtained as a the response to SQ or even assumed to be known). Indeed, by the definition of conditional probability, E[g(x,y) | f(x) = 1] = E[g(x,y)*f(x)] / E[f(x)]. However this means that to achieve tolerance \tau for the quantity we care about E[g(x,y) | f(x) = 1] one needs tolerance \tau * E[f(x)] when estimating E[g(x,y)* f(x)] in the classic SQ model. If E[f(x)] (which is the probability that the filter is satisfied) is small then tolerance required in classic implementation is much higher. For example, in the simple problem of learning thresholds we give on page 6 we need tolerance of 1/4 for E[g(x,y) | f(x) = 1] but E[f(x)] can be as low as the desired error \epsilon. To avoid such misunderstandings we will clarify this point in the final version as well.

Unfortunately, the rest of the review and the evaluation of our work are based on this misunderstanding of our model and its novelty.
The review also appears to entirely overlook the algorithmic contributions in the work: the first label efficient polynomial time algorithms for learning homogeneous linear separators in the presence of noise and implications to privacy in active learning.

Assigned_Reviewer_6 =======

Thank you for your positive comments and careful reading!

Assigned_Reviewer_8 ========

Thank you for your positive comments. We would like to briefly mention that the description of our work in your review is partially incorrect and oversimplifies our contribution. We do not give "a method for turning certain active learning algorithms into active learning statistical queries (SQ)" but rather present active SQ model and give new learning algorithms in this model. Algorithms in this model can then be automatically transformed to noise tolerant and differentially-private active learning algorithms. The active SQ algorithms that we describe are of two types: some build directly on ideas developed for active learning and SQ model (e.g. learning linear separators over logconcave distributions in Sec. 3) while other are based on new algorithmic ideas (e.g. learning linear separators over uniform distribution in Theorem 3.5 and learning of rectangles in Sec. 2.3).

We would also like to address "not surprising" which is mentioned in your comments about our contribution. Designing efficient noise tolerant algorithms has been one of the major questions in Active Learning with very little progress before our work. There are also no prior differentially-private active learning algorithms. Indeed, the initial idea of combining SQ model with active learning to achieve noise tolerance is a natural approach to the problem. It is however easy to overlook when presented directly with the finished result that a priori it is unclear if such a marriage is even possible. As we mention in the paper, being active and statistical appear to be incompatible requirements on the algorithm. Active algorithms typically make label query decisions on the basis of examining individual samples. At the same time statistical algorithms can only examine properties of the underlying distribution. The bulk of our contribution is to find the right definitions and to give new active learning algorithms based on these definitions. One of the algorithms we give differs substantially from the previously known ones.